METHODS AND RESOURCES

# An engineered bacterial symbiont allows noninvasive biosensing of the honey bee gut environment

**Audam Chhun** *, Silvia Moriano-Gutierrez, Florian Zoppi, Amélie Cabirol, Philipp Engel*, Yolanda Schaerli ⓘ *

Department of Fundamental Microbiology, University of Lausanne, Lausanne, Switzerland

* audam.chhun@unil.ch (AC); philipp.engel@unil.ch (PE); yolanda.schaerli@unil.ch (YS)

## Abstract

The honey bee is a powerful model system to probe host–gut microbiota interactions, and an important pollinator species for natural ecosystems and for agriculture. While bacterial biosensors can provide critical insight into the complex interplay occurring between a host and its associated microbiota, the lack of methods to noninvasively sample the gut content, and the limited genetic tools to engineer symbionts, have so far hindered their development in honey bees. Here, we built a versatile molecular tool kit to genetically modify symbionts and reported for the first time in the honey bee a technique to sample their feces. We reprogrammed the native bee gut bacterium *Snodgrassella alvi* as a biosensor for IPTG, with engineered cells that stably colonize the gut of honey bees and report exposure to the molecules in a dose-dependent manner through the expression of a fluorescent protein. We showed that fluorescence readout can be measured in the gut tissues or noninvasively in the feces. These tools and techniques will enable rapid building of engineered bacteria to answer fundamental questions in host–gut microbiota research.

## Introduction

Microbes inhabiting our gastrointestinal tract play pivotal roles in human health and physiology [1]. Gaining insight into host–microbiota interactions is, however, inherently arduous because the gut environment is not readily accessible as its content can only be queried by dissection of the gut tissue, invasive sampling methods, ingestible sampling devices, or indirectly by sampling fecal matter or analyzing the volatilome of the host [2–5]. Moreover, compounds in the gut will be metabolized by either the host or its associated microbes and may only appear at very specific times and locations within the gastrointestinal tract. As such, existing methods to study the microbiota often fail to capture these elusive dynamics by providing only snapshot views of limited resolution, which restricts progress of the field [6–8]. The development of tools to obtain spatial and/or temporal information in situ is hence much required to untangle the convoluted and dynamic interplay occurring within the microbiota and with its host. Synthetic biology holds promise to advance the field in this direction by engineering human

**Data Availability Statement:** The plasmids developed in this study have been deposited in Addgene and are available via the following accession numbers: pAC04 (#197400), pAC06

(#197401), pAC08 (#197402), pAC09 (#197403), pAC10 (#197404), pAC11 (#197405), pAC12 (#197406), pAC13 (#197407), pAC14 (#197408), pAC23 (#197409), pAC24 (#197410), pAC25 (#197411), pAC26 (#197412), pAC17V5a (#197413) and pAC17V5b (#197414). Source data are provided with this manuscript in the S1 Data file. The Fiji macro used for fluorescence quantification of cells present in fecal samples is provided as supplementary information and can be found in S1 Text.

**Funding:** This work was supported by the University of Lausanne, the NCCR Microbiomes (National Centre of Competence in Research), funded by the Swiss National Science Foundation (grant no. 180575) to P.E and Y.S. and the Marie Skłodowska-Curie fellowship HarmHoney (grant no. 892574) to A.Ca. The funders had no role in study design, data collection and analysis, decision to publish, or preparation of the manuscript.

**Competing interests:** The authors have declared that no competing interests exist.

**Abbreviations:** CFU, colony-forming unit; DAP, diaminopimelic acid; DAPI, 4,6-diamidino-2-phenylindole; GFP, green fluorescent protein; IPTG, isopropylthio-β-galactoside; LB, Lysogeny Broth; qPCR, quantitative PCR; TSB, Tryptic Soy Broth.

commensal bacteria as live biosensors [9–12]. For instance, a murine *Escherichia coli* strain was modified to detect tetrathionate (a transient by-product of inflammation) within the mouse gut, to "remember" and to report exposure to the molecule in the feces [13]. Although these novel biosensors have great potential to examine the gut environment, their utilization to deepen our knowledge of the microbiota remains challenging as these communities are typically extremely diverse and comprise many intractable species.

In comparison to the complex mammalian microbiota, the microbial community found in the gut of honey bees is remarkably simple in its composition [14]. Its main bacterial members are readily culturable in the laboratory, and microbiota-depleted individuals can easily be obtained [15,16]. As a result, honey bees represent a powerful model to improve our understanding of the underlying principles governing the assembly and function of gut microbial communities [17]. Furthermore, bees are key pollinators of many crops and wild flowers, and yet, their populations have been endangered by multiple anthropogenic stressors, including antibiotics [18–20]. Employing bacterial biosensors for the study of honey bees would offer considerable opportunities to expand our knowledge on the biology of their gut microbiota both under health and disease.

However, several challenges currently prevent their application for the bee gut environment. Reported bacterial biosensors have so far relied almost exclusively on strains of *E. coli*, which has restricted their use to a handful of hosts including *Caenorhabditis elegans* and mice [21–23]. Solely bona fide native symbionts of bees could colonize their guts with minimal alteration of the studied community and its environment [24]. Also, while the utilization of bacterial biosensors to examine the gut environment relies on feces sampling and subsequent analyses of the recovered bacteria, fecal analysis has never been applied to honey bees as they do not defecate in laboratory conditions [25]. More importantly, the microbial community residing in the gut of honey bees has been formally described only recently [26,27]. The first report of a functional molecular tool kit for these bacterial species was just published 5 years ago, and it was later remarkably used to reprogram a bee gut symbiont to promote health of its host, as well as to regulate bee gene expression by symbiont-mediated RNAi [28–31]. More recently, a new method based on homologous recombination allowed for one-step genomic knock-out and knock-in in a honey bee gut bacterium [32]. These studies brought to light the potential of microbial engineering for the field of honey bee gut microbiota, but the ensemble of genetic parts available to date is still limited. For example, the tool kit comprised a collection of plasmids based on the RSF1010 replicon [28]. Although effective, these vectors present shortcomings, as the size of the replicon itself (approximately 6 kb) complicates cloning. The lack of other functional replicons also prevents the use of multiple compatible plasmids in single bacterial cells, which can be advantageous for the construction of complex genetic circuits that require partitioning [9,33].

Here, we aimed at engineering a honey bee gut symbiont as the first biosensor of metabolites for the intestinal environment. As a proof of concept, we reprogrammed *Snodgrassella alvi*, a bacterium found in the proximal gut of honey bees, as a noninvasive diagnostic tool that senses and reports the presence of the sugar derivative isopropylthio-β-galactoside (IPTG) in situ by expressing a fluorescent reporter. For this, we developed a method to collect bee feces and pioneered its use for the longitudinal characterization of engineered cells stability in vivo. We also expanded the set of genetic tools dedicated to native honey bee gut symbionts through the development of a collection of broad-host range replicative plasmids that can be stably maintained in *S. alvi*, as well as an IPTG-inducible promoter to genetically manipulate this nonmodel bacterial species. Engineered cells' response to the presence of IPTG was dose dependent and could be determined both from the gut tissue and from the bee feces.

## Results

### Noninvasive gut microbiota sampling via feces collection

We first developed an electroporation protocol to routinely transform DNA into *S. alvi* (see Methods section), as a faster and simpler technique than the conjugal transfer typically employed for this species [28]. We were able to electroporate the previously described plasmid pBTK570 to wild-type cells [28], resulting in spectinomycin-resistant *S. alvi*. Next, we fed microbiota-depleted honey bees with an inoculum of engineered cells and allowed for colonization of the gut. We then established a noninvasive protocol to repeatedly collect honey bee feces and analyze its bacterial composition over time (Fig 1). By applying a gentle pressure onto the abdomen of $CO_2$-stunned bees, a substantial amount of fecal matter could be retrieved (Fig 1A). The volume of feces averaged 4.4 ± 2.4 µl per bee (*n* = 18), with a 90% success rate of extraction (Fig 1B). To confirm that the collected feces allow recovery of engineered bacteria and that they are an informative proxy for the honey bee gut bacterial content, we looked for the presence of the engineered strain of *S. alvi* in the feces and guts (i.e., midgut, ileum, and rectum) of inoculated bees (Fig 1C).

For this, feces were collected a week postinoculation, directly followed by dissection of the gut tissues of those same individuals. The concentration of engineered *S. alvi* present in the feces and gut samples was estimated by isolating bacterial colonies onto selective media. Interestingly, genetically modified bacteria were found in the feces, with an average bacterial load of $1.3 \times 10^{-5}$ colony-forming unit (CFU) per µl of feces (Fig 1C). It shows that gut symbionts commonly colonizing the proximal gut can nonetheless be abundantly found in fecal matter. Furthermore, we found a positive correlation between the amount of *S. alvi* in the feces versus the gut (Pearson correlation coefficient R = 0.3963, *p*-value < 0.05; S1 Fig). Not only were bacterial loads correlated, but absence of bacteria in feces indicated noncolonized bees (i.e., no bacteria in gut tissues; Fig 1C). Altogether, our data suggest that honey bee feces are a reliable indicator of the level of bacterial gut colonization.

Lastly, we checked that the feces extraction procedure established could be performed without being detrimental to the bees' health. Results showed that weekly feces collection did not significantly affect the probability of survival of manipulated honey bees compared to the non-treated individuals (Fig 1D). Being now able to noninvasively probe the gut content of honey bees via fecal extraction allows for their longitudinal study as well as for the use of bacterial biosensors.

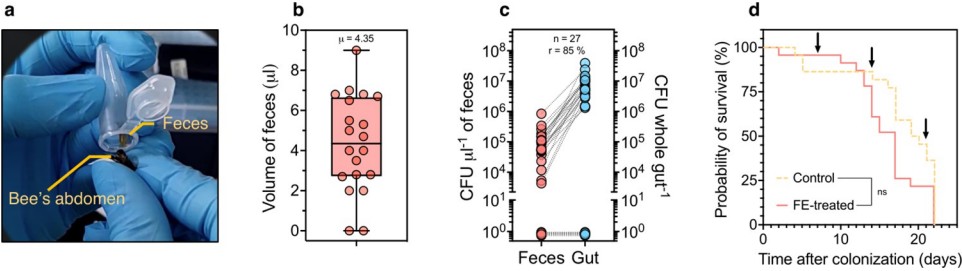

**Fig 1. A novel method for noninvasive sampling of the honey bee gut. (a)** Photograph of a bee feces extraction. **(b)** Box plot indicates the median volume of extracted feces from *n* = 18 bees. **(c)** Dot plot shows bacterial load of engineered *S. alvi* found in the feces and in the gut homogenates. Dotted lines between data points indicate matching samples (i.e., feces and gut were sourced from the same bee). Number of bees sampled (*n*) and the rate of colonization (*r*) for the experiment are indicated. **(d)** Feces collection does not significantly decrease bees fitness. Kaplan–Meier graph shows honeybee's probability of survival over time when subjected to fecal extractions (FE-treated; solid line) compared to control bees (Control; dotted line). Times of feces collection are indicated by arrows. Log-rank test, not significant (ns) at *p*-value = 0.112 for *n* = 44 bees. The data underlying this Figure can be found in the S1 Data file, sheets "Fig 1B," "Fig 1C," and "Fig 1D".

**Table 1. Main features of the broad-host range replicons tested in *S.alvi* and *B. apis*.**

| Broad-host range replicon | RK2 | pBBR1 | pVS1 | RSF1010 | pTF-FC2 |
|---|---|---|---|---|---|
| Size (kb) | 2.4 | 2.6 | 3.8 (6.1)* | 5.7 | 6.2 |
| Original host | *Pseudomonas aeruginosa* | *Bordetella bronchiseptica* | *Pseudomonas aeruginosa* | *Escherichia coli* | *Thiobacillus ferrooxidans* |
| References | • Thomas (1982)[34]<br>• Wirth (2020)[35] | • Antoine (1992)[36]<br>• Prior (2010)[37] | • Itoh (1984)[38]<br>• Heeb (2000)[39] | • Guerry (1974)[59]<br>• Leonard (2018)[28] | • Mao (1980)[40]<br>• Rawlings (1984)[41] |
| Stably maintained in *S. alvi* | No | Yes | Yes | Yes | Yes |
| Stably maintained in *B. apis* | Yes | Yes | No | Yes | Yes |

*pVS1 does not replicate in *E. coli*. The p15A replicon was therefore previously added to allow cloning in *E. coli*. The overall size of the two replicons is indicated between parentheses.

### Functional broad-host range plasmids for honey bee symbionts

To expand the range of molecular tools available to genetically manipulate honey bee gut bacteria, we constructed broad-host range plasmids that can stably replicate in *S. alvi*. While the RSF1010 replicon had previously been shown to propagate in honey bee gut symbionts [28], we identified in the literature other candidate broad-host range replicons that could also potentially do so, namely, RK2 [34,35], pBBR1 [36,37], pVS1 [38,39], and pTF-FC2 [40,41] (**Table 1**). They were selected based on the range of species in which they were shown to function and their small sizes. Each origin of replication was cloned in standardized vectors carrying distinct combinations of fluorescent proteins and antibiotic resistance markers (i.e., E2-crimson/specR, GFP/ampR, or E2-crimson/ampR), resulting in a collection of 11 broad-host range plasmids (**Figs 2 and S2**).

Upon electroporation of the plasmids in *S. alvi*, we were able to obtain stable cell lines from all replicons tested, except for RK2, confirming their ability to replicate in this species (**Table 1**). Additionally, we considered whether our broad-host range plasmids could be employed for other members of the honey bee gut community. As preliminary evidence, we transformed the vectors in *Bartonella apis*, a proteobacterium from a different class than *S. alvi* [27], and found that RK2, pBBR1, and pTF-FC2 stably replicate in *B. apis* (**Table 1**).

We further characterized each replicon, alongside RSF1010, by estimating their copy numbers, protein expression levels and co-compatibility in *S. alvi* (**Fig 3**). Data collected by

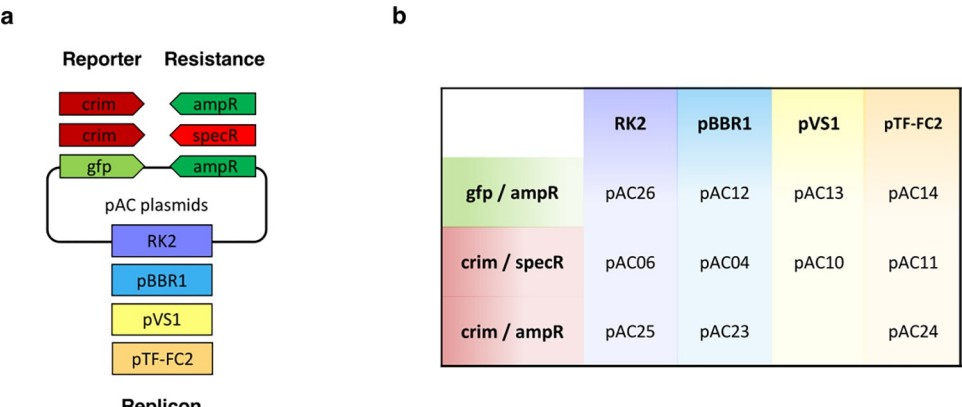

**Fig 2. Collection of broad-host range plasmids developed in this study. (a)** Schematic showing the different available combinations of origin of replications and standardized fragments bearing the antibiotic marker (ampR, specR) and fluorescent reporter (crim, gfp). **(b)** The names of the corresponding plasmids are indicated. Detailed plasmid maps can be found in **S2 Fig**.

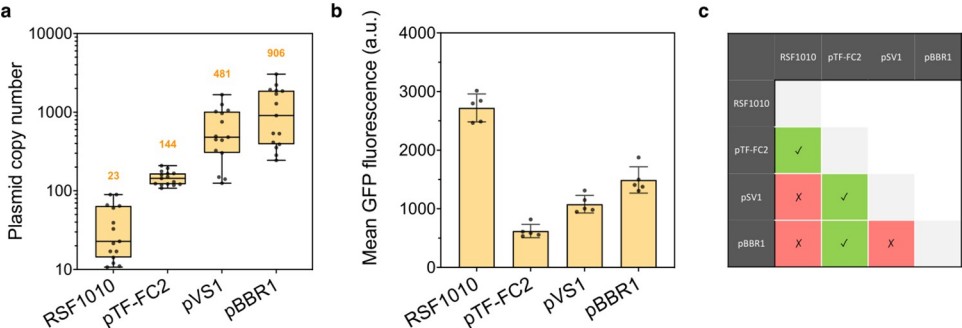

**Fig 3. Characterization of functional broad-host range replicons in the honey bee gut symbiont *S. alvi*. (a)** Broad-host range plasmids have different copy numbers in *S. alvi*. Box plots show median values of plasmid copy numbers obtained by qPCR from 3 independent experiments with 5 biological replicates each (total *n* = 15). Median copy numbers are indicated above the corresponding box plots. **(b)** The difference in plasmid copy number results in different protein expression levels in *S. alvi*. Graph shows mean of GFP fluorescence ± standard deviations of 5 biological replicates. Each replicate represents the average fluorescence of at least 9,000 cells measured by flow cytometry. Plasmids used for panels **a** and **b** in *S. alvi* were pAC08, pAC14, pAC13, and pAC12, carrying the RSF1010, pTF-FC2, pVS1, and pBBR1 origins of replication, respectively. **(c)** Some replicons are compatible and can be cotransformed in *S. alvi*. Matrix table indicates compatible (green boxes with check mark) and incompatible (red boxes with cross mark) replicons. Vectors were found compatible upon their successful cotransformation by electroporation in *S. alvi*. The data underlying this Figure can be found in the **S1 Data** file, sheets "Fig 3A" and "Fig 3B".

quantitative PCR (qPCR; **S3 Fig**) in *S. alvi* showed that the vectors propagate to varying copy numbers, ranging from medium copy (approximately 23 copies per cell for RSF1010) to high copy number (approximately 906 copies per cell for pBBR1; **Fig 3A**). The fluorescence of individual engineered cells carrying the different replicons was also assessed by flow cytometry (**Fig 3B**). Except for *S. alvi* bearing the RSF1010 origin of replication, which intriguingly exhibited the highest fluorescent signal while harboring in principle the lowest amount of plasmid, cells carrying the other replicons produced levels of fluorescence consistent with the determined plasmid copy number (**Fig 3A and 3B**). These differences translated into a moderate dynamic range, with up to a 4-fold increase in fluorescence between the highest and lowest average green fluorescent protein (GFP) signals (i.e., pTF-FC2 and RSF1010; **Fig 3B**). What is more, by cotransforming in pairs the different plasmids in *S. alvi*, we observed that the pTF-FC2 replicon was compatible with all other origins of replication tested and allowed stable dual vector propagation (**Fig 3C**). The same analyses were also carried out in *B. apis* and led to largely similar results (**S4 Fig**). Taken together, characterization of the plasmid collection illustrated its versatility, as it offers a variety of co-compatible expression vectors with distinct properties, thus allowing distribution of genetic circuits and fine-tuning of their output.

## Gut colonization stability over time of engineered *S. alvi*

Having expanded the set of plasmids to engineer honey bee gut symbionts, we validated the maintenance of such plasmids in vitro, as well as in vivo when engineered cells are in the presence of the natural gut microbial community, and also in absence of antibiotics that may disrupt the natural microbiota (**Fig 4**). For this, we chose to test the set of vectors providing spectinomycin resistance (crimson/specR), as they prevent the growth of nonresistant satellite colonies typically obtained using ampicillin selection upon CFU estimation. We first performed in vitro experiments where *S. alvi* bearing the different corresponding broad-host range vectors were grown in liquid cultures with or without antibiotic selection. Upon reaching stationary phase, cells from both conditions were isolated onto nonselective solid media,

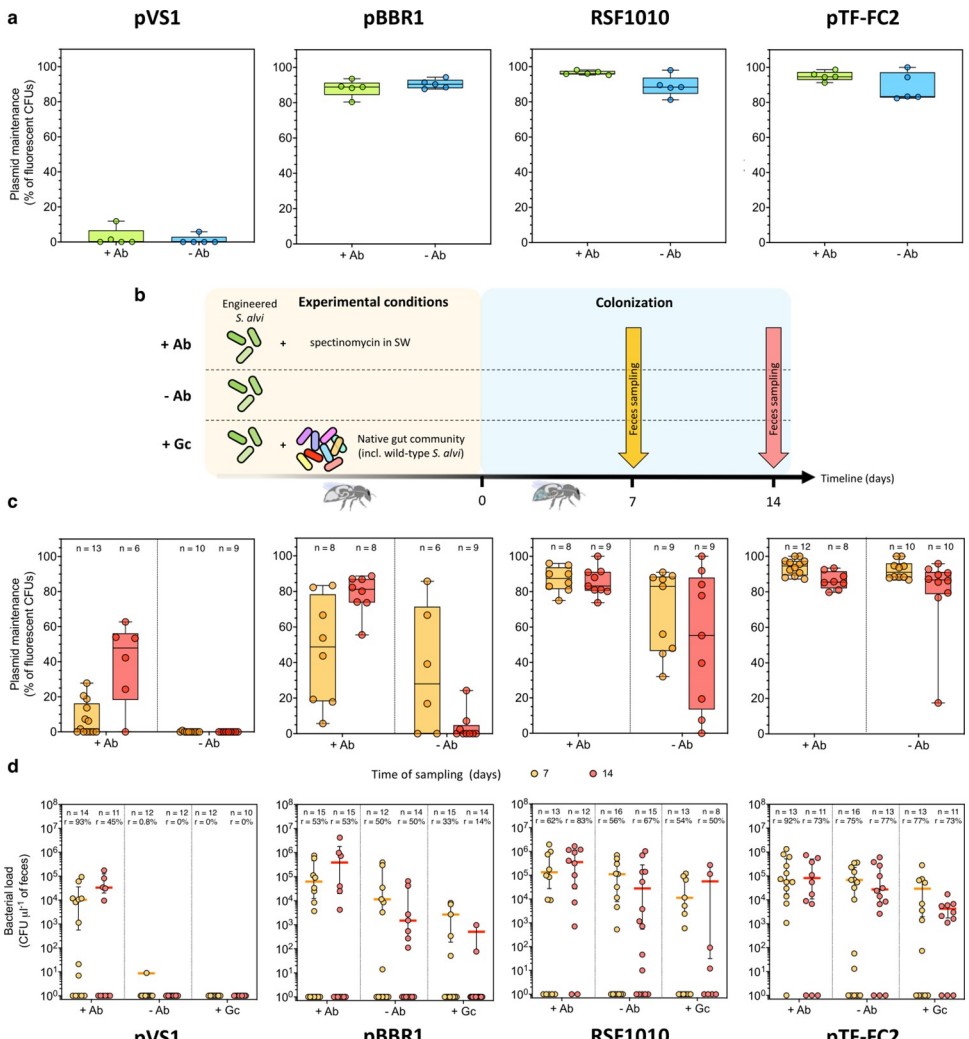

**Fig 4. Maintenance of broad-host range plasmids and engineered *S. alvi* over time.** Each broad-host-range vector has a distinct level of maintenance in *S. alvi* cells, as demonstrated in vitro **(a)** and in vivo **(c)**. Box plots represent plasmid maintenance median values of *n* biological replicates. Plasmid maintenance was estimated by measuring the percentage of fluorescent colony forming units (CFUs) plated onto nonselective media of samples obtained from **(a)** liquid cultures (in vitro) after 3 days of growth with (+Ab) or without (−Ab) spectinomycin supplementation; or **(c)** from feces (in vivo) collected 7 (orange) and 14 (red) days after gut monocolonization of bees continuously fed sugar water supplemented with (+Ab) or without (−Ab) spectinomycin. Only samples for which we detected bacterial colonies upon plating the fecal material were considered. **(b)** Schematic outline of the in vivo experiment to estimate engineered *S. alvi* colonization maintenance. Arrows indicate the 2 times of feces sampling. **(d)** Bacterial load of engineered *S. alvi* cells found in feces collected 7 (orange) and 14 (red) days after gut monocolonization of bees continuously fed sugar water supplemented with (+Ab) or without (−Ab) spectinomycin. Concentrations of our different *S. alvi* strains isolated from the feces of bees co-colonized with the natural gut community and that were fed sugar water without spectinomycin (+Gc) was also analyzed. Those CFUs values were obtained by plating diluted feces onto selective media (i.e., supplemented with spectinomycin). Bees for which we did not detect engineered bacteria were considered noncolonized individuals. The rate of colonized bees (*r*) and the number of bees sampled (*n*) are provided. Corresponding median values of colonized bees only are represented by colored horizontal bars with interquartile ranges. Plasmids used for all panels in *S. alvi* were pAC10, pAC04, pBTK570, and pAC11, carrying the pVS1, pBBR1, RSF1010, and pTF-FC2 origins of replication, respectively. The data underlying this Figure can be found in the **S1 Data** file, sheets "Fig 4A," "Fig 4C," and "Fig 4D".

allowing bacteria with and without plasmid to grow, and the ratio of fluorescently labeled bacterial colonies was measured as a proxy for plasmid maintenance (**Fig 4A**). Samples across conditions were also plated onto selective media and all bacteria exhibited fluorescence, confirming that observed unlabeled cells growing on nonselective media were the result of plasmid loss rather than mutations that would lead to the loss or inactivation of the fluorophore. Ultimately, the pBRR1, RSF1010, and pTF-FC2 replicons were found stably maintained within the experimental timeframe, as the average percentage of cells carrying the vectors when incubated without selection pressure remained at approximately 89%. On the contrary, the propagation of pVS1 appeared remarkably weak, with less than 3% of the bacterial population still bearing the plasmid at the end of the experiment, and thus even for the control condition comprising cells grown with antibiotics. We hypothesize that this was due to an artifact of the experimental setup where plasmid stability is so low that although all single cells incubated with antibiotics carry the vector when spotted onto nonselective media, the resulting macrocolonies show no fluorescence as bacteria rapidly lose the plasmid throughout cell division. Low maintenance values reported for the "+Ab" condition (i.e., <80%) are thus not accurate in absolute terms but remain an indicator of very feeble plasmid propagation.

Additionally, we characterized plasmid maintenance in vivo by monocolonizing 4 groups of microbiota-depleted honey bees with *S. alvi* bearing the different broad-host range plasmids. Each group of bees was divided into 2 subgroups that were fed either normal (−Ab) or antibiotic-supplemented (+Ab) sugar water. We then carried out a longitudinal analysis of their microbiota by weekly feces sampling over 14 days (**Fig 4B**). Plasmid maintenance for each replicon was calculated similarly to what had been done for the in vitro samples, using cells isolated from the collected fecal matter (**Fig 4C**). Consistent with the in vitro data, the pVS1-based vector propagated poorly in vivo, as the entire *S. alvi* population recovered from nontreated bees was no longer carrying the plasmid a week postinoculation. Interestingly, at the scale of the average life span of caged honey bees (i.e., approximately 3 weeks) [42], the other replicons displayed distinct stability profiles ranging from short to long term, even without the presence of antibiotics. For instance, the pBBR1-based plasmid was lost quickly, with a median of 28% of the bacterial population bearing the plasmid after a week, before dramatically decreasing to 0% after 14 days. The 2 abovementioned vectors that are rapidly lost have the highest copy numbers in *S. alvi* (**Fig 3A**), which might indicate that the metabolic burden resulting from their replication significantly impacts their maintenance within cells. Contrastingly, the lower-copy RSF1010- and pTF-FC2-based vectors showed robust propagation over time, with a respective median of 55% and 86% of the bacterial populations still carrying the plasmids after a 2-week period without selection pressure. The pTF-FC2 plasmid is known to encode for a maintenance system, which might participate in its steady propagation in our cells (see Discussion).

Finally, due to the potential fitness cost linked to the replication of heterologous genetic material, we wondered whether utilization of the broad-host range plasmids would affect the ability of the engineered cells to colonize and persist in the honey bee gut over time in the presence of the natural microbiota. To answer this, we coinoculated microbiota-depleted honey bees with modified *S. alvi* strains harboring the different vectors and the natural gut community (+Gc), which included wild-type *S. alvi* (**Fig 4B**). Feces were sampled from these colonized bees and analyzed together with those collected for the measurement of plasmid stability. Fecal samples from the 3 treatment groups (i.e., +Ab, −Ab, and +Gc) were ultimately examined by isolating engineered *S. alvi* onto selective solid media, allowing us to determine the levels of gut colonization through time (**Fig 4D**). In the case of *S. alvi* transformed with the pVS1-based plasmid, they were detected consistently over 14 days at a concentration of approximately $10^4$ cells per μl of feces but solely from bees that were fed antibiotic. This further suggests that low

plasmid maintenance values for the "+Ab" condition in Fig 4A are indeed due to an artifact of the experimental setup and that maintaining the pVS1 origin of replication might be detrimental to cells' fitness and that stable colonization of the gut requires the addition of selection pressure. On the other hand, *S. alvi* bearing the pBBR1, RSF1010, and pTF-FC2 replicons were able to establish persisting communities that were not constrained to the presence of antibiotics, with the detection of approximately $10^3$ to $10^5$ engineered cells per μl of feces across the experiment. Bacterial concentrations and success rates of colonization found for each strain were also in line with the varying degrees of plasmid maintenance established in this study. *S. alvi* carrying the pTF-FC2-based plasmid, for instance, reached median values of 2.7 $10^4$ cells per μl of feces after 2 weeks, while the less frequently maintained vector pBBR1 resulted in fecal bacterial populations 18 times smaller. These observations were also consistent with the rates of nonantibiotic-treated honey bees successfully colonized by each strain, as on average 50%, 61%, and 76% of bees were colonized by *S. alvi* transformed with the pBBR1, RSF1010, and pTF-FC2 replicons, respectively.

More importantly, the *S. alvi* strains genetically modified with the broad-host range vectors, except for pVS1, were able to colonize the honey bee gut environment upon competition with the native gut community, albeit to different degrees of success (**Fig 4D**). Cells bearing the RSF1010 and pTF-FC2 plasmids, for example, were both detected at concentrations of approximately $10^4$ cells per μl of feces across sampling times, but they differed in their average colonization rates, which were 52% and 75%, respectively. Nevertheless, the metabolic burden imposed by the broad-host range plasmids allowed engineered cells to settle as part of the bee microbiota without being outcompeted by their wild-type counterparts. This is of relevance as it confirms their potential to be utilized in situ for biosensing of the gut environment and its microbial communities.

## Reprogrammed *S. alvi* cells act as in situ biosensors for IPTG

Engineering of bacterial biosensors relies on promoters that are specifically induced upon exposure to molecules of interest and that subsequently drive the expression of dedicated reporter genes. To date, the only functional inducible system reported for members of the honey bee gut microbiota is based on the previously described pBTK552 vector, which carries a fluorescent protein whose expression is conditional to the presence of the widely used sugar derivative IPTG [28]. The efficacy of induction was, however, solely demonstrated in vitro. Additionally, upon its transformation in *S. alvi*, we were unable to maintain integrity of the plasmid in our experimental conditions, as the system showed high genetic instability resulting in frequent deletion of the *gfp* gene (**S5 Fig**). Consequently, we built several variants of an IPTG biosensor using our tool kit in different gene arrangements and obtained 3 stable genetic constructs. Two were based on a dual-plasmid system whereby the circuit was distributed between 2 compatible and stably maintained broad-host range plasmids (i.e., RSF1010 and pTF-FC2; **S6A Fig**). Following the characterization of the different constructs, we selected the pAC17V5 circuit to pursue our work, as it showed the best dynamic range in response to IPTG (**S6B Fig**). Briefly, the pAC17V5 system consists of a first pTF-FC2-based plasmid bearing the *lacI* gene constitutively expressed (pAC17V5b) and a second RSF1010-based vector carrying *gfp* under the control of a constitutive promoter with a Lac operator (lacO) sequence directly downstream (pAC17V5a; **Fig 5A**). When transformed together in *S. alvi*, the plasmids enabled cells to respond in vitro to IPTG with a 7-fold increase in GFP expression (**Fig 5B**). Additionally, we wondered if other species of the honey bee gut microbiota could also be engineered to respond to IPTG. We tested our 3 genetically stable constructs in the 2 γ-Proteobacteria *Gilliamella apicola* wkB7 and *Gilliamella apis* ESL0169 from the honey bee, as well as in

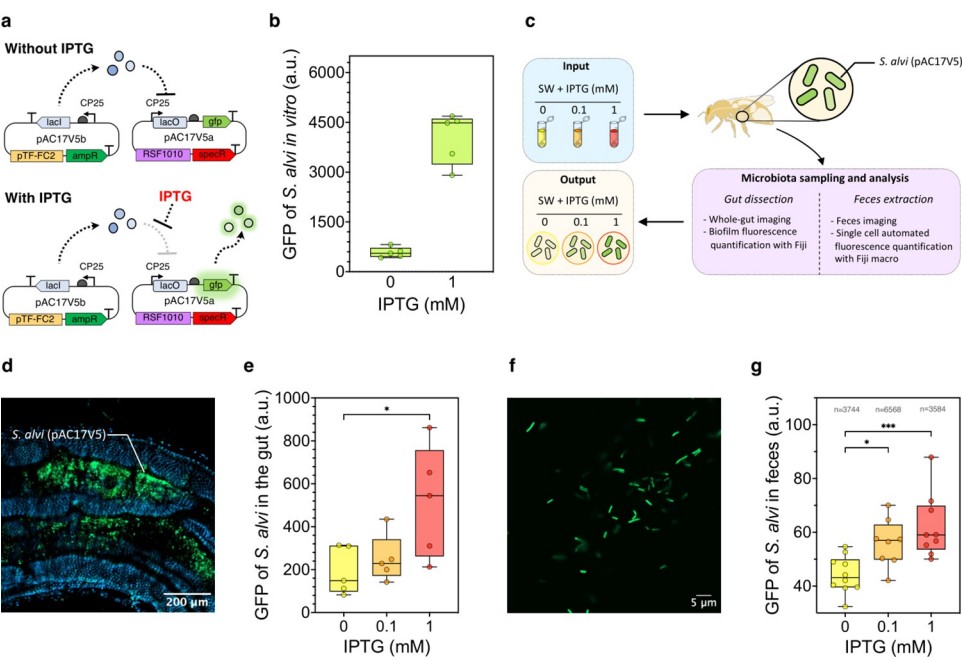

**Fig 5. Engineered *S. alvi* sense IPTG in vivo and report exposure in the gut tissue and feces.** (a) Schematic of the IPTG-inducible dual plasmid system pAC17V5. (b) *S. alvi* cells engineered with pAC17V5 respond to IPTG exposure in vitro. Box plots represent median values of GFP fluorescence of 5 biological replicates. Each replicate value is based on the average fluorescence of at least 9,000 *S. alvi* cells measured by flow cytometry, which were grown in liquid with (+) or without (−) IPTG. (c) Schematic outline of the in vivo experiment using reprogrammed *S. alvi* to sense IPTG in situ. (d) Confocal microscopy image of a bee gut colonized with engineered *S. alvi* cells bearing pAC17V5. A biofilm of *S. alvi* within a gut crypt is shown. The bee was fed with sugar water supplemented with 1 mM IPTG. Blue channel shows DAPI staining of host and bacterial cells. Green channel shows GFP fluorescence. (e) *S. alvi* engineered with pAC17V5 display a dose-response to IPTG exposure in vivo, which can be measured from gut tissue. Graph shows box plots representing median value of GFP fluorescence of *S. alvi* biofilms imaged from the gut of bees fed sugar water supplemented with either 0, 0.1, or 1 mM IPTG. Five bees were analyzed for each condition, and fluorescence values were averaged from 3 distinct sections of each gut. Tukey HSD test, * significant at $q$-value < 0.05. (f) Confocal microscopy image of bee feces, collected from an individual fed with sugar water supplemented with 1 mM IPTG and colonized with *S. alvi* cells bearing pAC17V5. Green channel shows GFP fluorescence. (g) *S. alvi* engineered with pAC17V5 display a dose-response to IPTG exposure in vivo, which can be measured from the feces. Box plots represent median values of GFP fluorescence of individual *S. alvi* cells imaged from the feces of bees fed sugar water supplemented with either 0, 0.1, or 1 mM IPTG. At least 8 bees were analyzed for each condition. The cumulated number $n$ of cells analyzed per conditions is indicated. Tukey HSD test, * significant at $q$-value < 0.05 and *** significant at $q$-value < 0.001. The data underlying this Figure can be found in the **S1 Data** file, sheets "Fig 5B," "Fig 5E," and "Fig 5G".

another strain of *S. alvi* isolated from stingless bees (**S7 Fig**). We showed that all 3 bacteria engineered with our one-plasmid system pAC17V3 (**S6A Fig**) were able to respond to the presence of IPTG in vitro (**S7 Fig**).

We then wished to establish whether the genetically reprogrammed *S. alvi* could also sense and respond to IPTG within the gut environment. For this, we monocolonized microbiota-depleted honey bees with *S. alvi* bearing the pAC17V5 dual-plasmid system, before splitting them into 3 groups that were fed sugar water supplemented with either 0, 0.1, or 1 mM IPTG. Feces and gut tissues were then collected from the honey bees, and the fluorescence of *S. alvi* cells present in each sample type was analyzed (**Fig 5C**). In whole guts, *S. alvi* was found to form biofilms localized in the intestinal crypts of the ileum (**Fig 5D**). Subsequent image analysis of those biofilms showed that engineered cells exhibited a dose-response to IPTG, whereby the strength of the measured fluorescent signals rose with increasing IPTG concentrations present in the food of the colonized honey bees (**Figs 5E and S8**).

Lastly, genetically modified *S. alvi* could also be monitored from their planktonic form contained in fecal material (**Fig 5F**). The quantification of cell fluorescence from confocal microscopy pictures of the feces was automated via a Fiji macro developed for this study (**S1 Text**), which allowed the analysis of thousands of individual bacteria in each condition. The obtained data further indicated that *S. alvi* could sense different concentrations of IPTG in the gut and respond accordingly with a dose-dependent fluorescent output (**Fig 5G**). Taken together, the in vivo experiments carried out confirmed that the native honey bee gut symbiont *S. alvi* can be genetically reprogrammed to act as an in situ biosensor of the intestinal environment. Moreover, the cell's response informing on the exposure to the detected molecule can be assessed locally, directly in the gut tissues, and noninvasively in honey bee feces.

## Discussion

We demonstrate here that the native honey bee gut bacterium *S. alvi* can be genetically reprogrammed as a biosensor of the intestinal environment, reporting exposure in a dose-dependent manner via the expression of a fluorescent protein. For this, we developed a collection of broad-host range plasmids, with diversity in maintenance and compatibilities, that expands the available genetic toolbox [28] and will facilitate further bacterial engineering in the field of honey bee microbiota. We showed that the reprogrammed symbionts can stably colonize the gut, even in the presence of the native community, and their fluorescent output can be directly recorded from the gut tissues. Furthermore, we report for the first time in the honey bee a noninvasive method to collect bee feces, allowing repeated monitoring of the fluorescence readout from the engineered symbionts that inform on the gut content. Altogether, our work paves the way for longitudinal studies of symbiont dynamics, as well as for the study of the physicochemical properties of the gut in its different microenvironments.

A striking advantage of bacterial biosensors is that they allow to noninvasively and nondestructively examine the gut content and its associated bacterial communities. In less than a decade, the development of technologies to reprogram human commensals like *E. coli* as biosensors has seen exciting progress, starting from the use of transcriptional regulator-based genetic circuits to sense simple chemicals [9], to the application of CRISPR to make bacteria record in their genomes transcriptional changes occurring as they pass through the guts [43]. For instance, it was shown that the human commensal *Bacteroides thetaiotaomicron* could be genetically reprogrammed to sense and respond to arabinogalactan supplemented in the food of colonized mice [10]. Similarly, the probiotic *E. coli* Nissle 1917 was engineered to simultaneously sense thiosulfate and nitrate in the gastrointestinal tract of mice as biomarkers for gut inflammation [11]. While the honey bee gut microbiota is an exceptional model to study host–microbe interactions [17], the development of bacterial biosensors dedicated to this system had yet to be undertaken.

Our research provides a collection of broad-host range plasmids that could replicate stably in the 2 prevalent honey bee symbionts *S. alvi* and *B. apis* (**Fig 2 and Table 1**). They were pivotal to the successful engineering of *S. alvi* as a biosensor. Indeed, we observed that several genetic constructs suffered from high genetic instability in this species (while being stable in *E. coli*), resulting in a loss of function of the circuit (for instance, pBTK503 and pBTK552; **S5 Fig**). The design that led to a stable and functional IPTG-inducible promoter in *S. alvi* relied on the partitioning of the circuit onto 2 compatible plasmids that were characterized in this study. More generally, the advent of sequencing technologies has revealed a plethora of new undomesticated bacterial species isolated from various environments, for which there is currently no known replicative plasmid to genetically manipulate them [44,45]. In this regard, our collection of broad-host range plasmids represents a useful resource of standardized vectors that can easily be tested in any nongenetically tractable species of interest.

Additionally, the broad-host range vectors displayed versatility in their maintenance within cells over time, which could be utilized for distinct applications. For instance, the pVS1-based plasmid propagated very poorly in *S. alvi* under nonselective condition (**Fig 4**). As pVS1 is not actually functional in *E. coli*, making traditional cloning work impossible, the p15A replicon was previously added to the plasmid so that it effectively carried 2 origins of replication [39]. We hypothesize that the presence of these 2 replicons taken together with the high copy number of the plasmid (i.e., approximately 500 copies per cell) may contribute to its remarkably quick loss. This is of interest for the potential application of transgenic organisms where the maintenance of modified genetic material shall remain transient. It has been previously reported, for example, that *S. alvi* could be engineered as a probiotic to promote honey bee resistance to deformed wing virus and *Varroa* mites [29]. This genetically modified bacterium has great potential for the improvement of colony health, but its application requires the development of robust biocontainment methods to prevent its spread into the environment. This remains a central concept in the design of any genetically modified microbes that are aimed to be used outside confined laboratory conditions, where strains are engineered with kill switches or to become auxotrophs so that they will not survive beyond their intended use [21,46,47]. The rapid loss of the pVS1 plasmid without antibiotic selection may thus be a valuable feature that could be repurposed so that the engineered cells only transiently carry transgenic information. On the other hand, some applications of genetically reprogrammed bacteria, like biosensing, require long-term functionality of the genetic circuits, which has been shown to be challenging in certain systems. For instance, up to 65% of engineered *L. paracasei* bacteria lost their recombinant plasmids during their passage through the rat gut [48]. Some of our broad-host range plasmids like pTF-FC2, conversely, were proven to be maintained stably in *S. alvi* and could propagate within the cell population for up to 2 weeks in vivo (**Fig 4C**). Those engineered cells were able to robustly colonize the honey bee gut and persist while competing with the natural microbiota (**Fig 4D**), highlighting the value of our broad-host range plasmid collection for in vivo applications. The reported maintenance levels of our plasmids, however, might be specific to the gene tested here (i.e., E2-crimson), and these values might differ when expressing genes that cause a higher metabolic burden or are toxic to the cells. Furthermore, potential horizontal gene transfer occurring between our modified *S. alvi* and wild-type cells in the +Gc experimental conditions could have inflated the bacterial loads reported. However, we believe the effect of spontaneous conjugation on the reported values to be minimal, as all bacterial concentrations measured are consistent with the levels of plasmid maintenance over time determined in vitro and in vivo with the +/−Ab conditions. Interestingly, the low-copy number pTF-FC2 vector has a toxin/antitoxin system naturally encoded within the replicon itself that surely contributes to this lasting plasmid maintenance [49]. Based on 3 genes of small sizes (approximately 700 bp), the system could easily be ported to other plasmids and could therefore potentially modulate their propagation and maintenance.

To provide a longitudinal readout of the colonization dynamics, as well as to monitor the stability of the constructed plasmids and engineered symbionts in vivo, we had to first develop a method to noninvasively and repeatedly collect the feces of honey bees (**Fig 1**). It was initially unclear whether engineered *S. alvi* cells could be isolated from feces and if those would reliably reflect the levels of gut colonization. Indeed, symbionts occupy distinct niches in the gastrointestinal tract, with species like *S. alvi* known to form biofilm in the proximal gut (i.e., ileum), while others are predominantly found in the distal gut (i.e., rectum) where fecal matter is retained [50]. Here, we showed that live *S. alvi* could be consistently isolated in large concentrations in the feces of honey bees over time (**Fig 4**), which suggests that bacteria typically settling in the proximal gut also accumulate in the rectum, probably due to shedding of gut tissues [51]. This has since been confirmed by another study carried out by our group that

characterized the fecal microbiota of honey bees and showed that all major genera of the gut microbiota are also found in the feces [52]. Considering that caged honey bees do not defecate, it is likely that bacterial accumulation was exacerbated in the rectum of individuals reared in the laboratory. This is of relevance as all studies of the gut microbiota of caged bees have so far exclusively examined whole-gut tissues to determine levels of bacterial colonization in the distal gut, therefore preventing discrimination between actual colonizing microbes (i.e., forming persistent biofilms) and the ones simply passing through and/or surviving in planktonic form in the lumen. In future studies, considering the bacteria retained in the rectum of caged individuals by systematic analysis of their fecal bacterial content could thus promote a more definite understanding of gut colonization in honey bees. Overall, similarly to the mammalian gut microbiota [53–55], fecal sampling over time and subsequent analysis holds promises for the improvement of our aptitude to observe and understand temporal dynamics occurring within the microbiota of honey bees. Our experiments also suggest that repeated feces collection does not significantly impact the fitness of honey bees, as no statistically significant difference in survival between nontreated and treated bees was measured. However, we observed a trend in which treated individuals might die earlier than nontreated bees, most likely because of moderate stress caused by handling during feces collection. Therefore, further studies investigating the precise impact of feces collection on bee health might be warranted. Also, while we provide experimental proof that feces sampling can be an effective technique to monitor the gut microbiota of caged honey bees over time, further work should be carried out to assess the potential limitations of this method for longitudinal studies of wild or hive bees.

Lastly, we designed a genetically stable inducible system based on 2 compatible replicons. As a proof of concept, we utilized it to reprogram *S. alvi* to sense IPTG in situ, therefore providing the first evidence that a prevalent honey bee symbiont can be engineered to behave as a biosensor (**Fig 5**). The fold-change induction observed in our system was modest, however, and might not be practical for easy assessment of the target compound concentration. Further optimization of the dynamic range should therefore be undertaken for future biosensors of biologically relevant molecules. We also tested if the set of IPTG-inducible circuits we developed could be used to reprogram other symbionts, including strains from the genus *Gilliamella*, as well as another strain of *S. alvi* isolated from the microbiota of stingless bees (**S7 Fig**). While the dual-plasmid system pAC17V5 was unfortunately not stably maintained in these species with its current design, most likely because of replicon incompatibilities, the RSF1010-based plasmid pAC17V3 allowed cells to respond to IPTG in vitro. These preliminary data are promising for the generalization of symbiont engineering as biosensors for the honey bee gut environment. However, as the gut microbiota is comprised of distinct and not closely related species, it is evident that most genetic tools (including plasmid replicons) will not work universally in all its members. This is a testimony to the difficulty of genetically engineering these environmental undomesticated symbionts [56], and more work will be required for the generalization of biosensor development across gut bacterial species. More importantly, future research efforts should focus on the development of biosensors dedicated to more biologically relevant chemical signals, such as oxygen, pH, pesticides, or short-chain fatty acids, for instance. The distribution of most small molecules across the gut is, indeed, either extremely localized or transient. Here, using confocal microscopy imaging, we could determine the fluorescence levels of engineered cells within biofilms of whole-gut samples. By exploiting the capability of confocal microscopy to acquire serial optical sections of such biofilms, the use of a bacterial biosensor may similarly allow to detect differences in concentrations of any molecules of interest across the depth of the gut. Coupled with our new ability to repeatedly probe the gut content by fecal sampling, engineered biosensors thus represent an exciting prospect to gather spatial and temporal information of great resolution regarding the

distribution of chemical signals in the gastrointestinal tract of honey bees, thus helping to shed light on the complex interplay between symbionts and the intestinal environment.

## Conclusions

To our knowledge, our work paves the way for further engineering of native symbionts of the honey bee to inform on the physicochemical properties of their corresponding niches within the gastrointestinal tract. The diagnostic ability of a genetically reprogrammed bacterium could, in principle, be tailored to virtually any signal of interest if responsive bacterial promoters exist for it, and the use of such bacterial biosensors should improve our understanding of the honey bee and its gut microbiota.

## Materials and methods

### Bacterial strains and plasmids

The type strains *S. alvi* wkB2[T] (ATTC No. BAA-2449) and *B. apis* PEB0122[T] (DSM No. 29779) were routinely grown in Tryptic Soy Broth (TSB, Bacto BD) and Columbia Broth supplemented with 5% defibrinated sheep blood (CBA, Difco BD), respectively. Cells were incubated at $34°C$ in a microaerophilic environment inside a 5% $CO_2$ incubator, with orbital shaking (170 rpm) when appropriate. Additionally, *E. coli* NEB 5-α (New England Biolabs) was used for cloning, while the diaminopimelic acid (DAP)-auxotroph *E. coli* JKE201 strain [57] was employed for conjugation with *B. apis*. Both *E. coli* strains were cultured in normal atmosphere in Lysogeny Broth (LB) at $37°C$ with orbital shaking (220 rpm), when needed. Colony selection and plasmid maintenance was achieved by supplementing the appropriate culture media with 30 to 60 μg ml$^{-1}$ spectinomycin (*S. alvi*/*B. apis*—*E. coli*), 30 to 100 μg ml$^{-1}$ ampicillin (*S. alvi*/*B. apis*—*E. coli*), and 60 μg ml$^{-1}$ DAP, when required.

The vector pDR401 was generously given by Dr. Shelly Deane (Stellenbosch University, South Africa). The plasmid pME6012 was kindly provided by Prof. Jan van der Meer (Lausanne University, Switzerland). The vectors pSEVA1213S (Addgene plasmid No. 122095) and pBMTBX-2 (Addgene plasmid No. 26073) were gifts from Dr. Pablo Ivan Nikel and Prof. Ryan Gill, respectively. The plasmids pBTK552 (Addgene plasmid No. 110618), pBTK503 (Addgene plasmid No. 110616), and pBTK570 (Addgene plasmid No. 110615) were gifts from Prof. Jeffrey Barrick.

### Construction of broad-host range and inducible vectors

To build the broad-host range plasmids, we cloned the different replicons together with a standard fragment bearing either (a) an ampicillin resistance marker with a GFP driven by the CP25 promoter, (b) a spectinomycin resistance marker with a E2-crimson fluorescent protein driven by the PA3 promoter, or (c) an ampicillin resistance marker with a E2-crimson fluorescent protein driven by the PA3 promoter. Each replicon was thus effectively cloned as 3 versions: GFP/ampR, E2-crimson/specR, or E2-crimson/ampR. The latter was generated to allow for the cotransformation of plasmids in *B. apis* to assess replicon compatibility, as it was found that plasmids expressing *gfp* were not genetically stable in this species. The pVS1 replicon, which did not replicate in *B. apis*, was therefore not cloned with the E2-crimson/ampR fragment.

The fragments GFP/ampR, E2-crimson/specR, and E2-crimson/ampR were amplified from the pAC08, pBTK570, and pAC09 vectors, respectively, using the primers AC_09/10. The replicons RK2, pBBR1, pTF-FC2, and pVS1 were obtained from the plasmids pSEVA1213S, pBMTBX-2, pDR401, and pME6012, respectively. The replicons were PCR amplified as 1 or 2 fragments, using the primers AC_16/17 (pSEVA1213S), AC_11/12 (pBMTBX-2), AC_48/49 and AC_50/51 (pDR401), and AC_46/46 with AC_45/47 (pME6012). Similarly, pAC17V5a

was constructed by linearizing the pBTK552 plasmid in 2 fragments, using the primers AC_59/AC_114 and AC_60/AC_70, before assembling them back together, effectively removing the *lacI* gene. The pAC17V5b vector was built by amplifying the *lacI* gene from pBTK552 using the primers AC_115/AC_116 and cloning it with the pTF-FC2 origin of replication obtained as 2 fragments with the primers AC_36/AC_50 and AC_48/AC_49. Ultimately, all fragments were assembled by Gibson assembly cloning using the NEBuilder HiFi DNA Assembly kit following manufacturer guidelines. Details about cloning primers can be found in **S2 Table.** The resulting broad-host range and inducible plasmids are listed in **S1 Table**, and full plasmid maps are depicted in **S2 and S6** **Figs**.

## Electroporation

To prepare electrocompetent cells, a single colony of *S. alvi* was used to inoculate a 5-ml liquid culture of TSB media. Cells were grown to early stationary phase for 2 to 3 days at 34˚C with orbital shaking (170 rpm) in a 5% $CO_2$ incubator. The culture was then diluted 1:50 in 50 ml of fresh TSB and incubated again for 24 hours in identical conditions to allow cells to reach mid-exponential phase (OD ≈ 0.3). Cells were then washed twice with 1:1 (v/v) ice-cold sterile 10% glycerol solution. For this, cells were centrifuged at 2,916*g* for 10 minutes at 4˚C, and the subsequent pellet was resuspended in the glycerol solution. Washed cells were then resuspended in 50 µl of sterile 10% glycerol solution and used for one transformation. Cells were either transformed fresh or stored at −80˚C.

Electroporation of cells was performed with an Eppendorf Eporator at 2.5 kV (capacitance 10 µF with resistance 600 Ω) with 0.25 µg of plasmid DNA in 0.2 cm-gap electroporation cuvettes. Cells were resuspended in 1 ml of TSB and allowed to recover for 2 hours at 34˚C with orbital shaking (170 rpm) in a 5% $CO_2$ incubator. Recovered cells were centrifuged (2,916*g* for 5 minutes at room temperature) and resuspended in 100 µl of fresh TSB before plating onto selective TSA media. Plates were incubated at 34˚C with 5% $CO_2$ atmosphere. Colonies resulting from successfully transformed cells typically appeared after 3 to 4 days, and transformation efficiencies averaged $6.6 \times 10^3 \pm 1.92 \times 10^3$ CFU µg$^{-1}$ of DNA.

## Conjugation

Broad-host range plasmids were transferred to *B. apis* by conjugation. For this, recipients *B. apis* were obtained by streaking single colonies onto CBA solid media to grow small bacterial lawns after 3 to 4 days of incubation. Cells were gently scrapped off the plates using a sterile plastic loop and resuspended in 500 µl of fresh CBA. The donor strain *E. coli* JKE201 transformed with the relevant plasmid was grown overnight in 3 ml of LB. The liquid culture was then diluted 1:100 (v/v) into 5 ml of fresh LB. Media used up to this stage was supplemented with DAP and antibiotics, as appropriate. Cells were grown for about 4 hours to mid-exponential phase (OD = 0.4 to 0.6) and then washed 3 times with fresh LB to remove antibiotics. Donor cells were resuspended in 500 µl of LB media without antibiotics.

For cell mating, donor and recipient cells were mixed, pelleted by centrifugation at 2,916*g* for 5 minutes, resuspended in 50 µl of sterile LB broth, and, finally, aliquoted as 10 µl spots onto solid CBA media supplemented with DAP. Cells were incubated overnight at 34˚C in a microaerophilic environment inside a 5% $CO_2$ incubator to allow mating. To select for exconjugant, cells were scrapped off the plates using a plastic loop and resuspended into 1 ml of sterile CBA. Bacteria were then centrifuged at 2,916*g* for 5 minutes and homogenized in 100 µl of fresh CBA, to ensure the removal of residual DAP. Finally, cells were plated onto selective CBA plates, which were incubated at 34˚C with 5% $CO_2$ atmosphere. Colonies resulting from successful conjugation typically appeared after 3 to 4 days.

## Flow cytometry

Cell fluorescence measurements were performed using a LSR Fortessa Flow Cytometer (BD) instrument and the BD FACSDiva acquisition software (version 9.0, BD). Collected data were processed using the analysis software FlowJo (version 10.8.1). Single colonies of *S. alvi* and *B. apis* bearing the different broad-host range plasmids were inoculated into liquid broth and incubated as described above until cells reached late exponential phase. Samples were then diluted 1:10 in sterile PBS prior to flow cytometry analysis. Fluorescent cells were detected by a double discrimination gating strategy, as follows: The cell population was first gated on a FSC-H/SSC-H plot, and fluorescent cells were then discriminated on a FITC-H (ex. 488 nm–em. 494 nm)/SSC-H or AlexaFluor700-H (ex. 633/40 nm–em. 696 nm)/SSC-H plot, for GFP and E2-crimson analysis, respectively (**S9 Fig**). The reported average fluorescence signal for each sample was calculated based on the measured mean fluorescence values of at least 9,000 cells.

## Plasmid copy number measurements by qPCR

Genomic DNA and plasmids were extracted using the Fast Pure Bacteria DNA isolation mini kit (Vazyme) and the QIAprep Spin Miniprep Kit (Qiagen), respectively. DNA concentrations were measured with a Qubit device (Qubit 3.0) and a Qubit dsDNA BR Assay kit (Invitrogen). Standard curves were generated based on known concentrations of serial diluted *S. alvi* and *B. apis* genomic DNA, as well as a miniprep of the pBTK570 vector (**S3 Fig**). qPCR primers used to detect the broad-host range plasmids were AC_24/25, which amplify 100 bp from the *E2-crimson* gene. Primers for *S. alvi* and *B. apis* genomic DNA detection were AC_28/29 and AC_30/31, respectively. Details about the qPCR primers can be found in **S2 Table**. Each reaction was performed in triplicate, and no-template controls were added to every run to confirm that the reaction mixtures were not contaminated. The generation of specific PCR products by the qPCR primers was confirmed by melting curve analysis and via observation of amplicons on an agarose gel (**S3 Fig**). Generated standard curves were used to calculate the qPCR efficiency E (values comprised between 90% and 110%) and linearity $R^2$ of amplification (**S3 Fig**).

Plasmid copy number was determined from bacterial lysates. For this, single colonies of *S. alvi* and *B. apis* bearing the different broad-host range plasmids were inoculated into liquid broth and incubated as described above until cells reached late exponential phase. Of those cultures, 1 ml was placed into sterile 1.5 ml Eppendorf tubes, boiled for 10 minutes at 100°C in a dry block Thermomixer (Eppendorf), and stored at −20°C until sample processing. Obtained lysates were thawed on ice, homogenized by brief vortexing, and 1 μl of sample was used as template for qPCR reactions, which were performed in a total volume of 10 μl consisting of 5 μl of 2X SYBR Select Master mix (Thermo Fisher), 3.6 μl of MiliQ water, and 0.2 μl of each of the appropriate 10 μM primers. Amplifications were carried out with a QuantStudio 5 real-time PCR system (Thermo Fisher), with the following conditions: 2 minutes at 50°C, 2 minutes at 95°C, followed by 40 cycles of 15 seconds at 95°C, and 1 minute at 60°C. Copy number per cell for each plasmid was calculated by dividing quantities of plasmids with the amount of genomic DNA for each sample, which were determined from the generated standard curves. Copies of genomic DNA were normalized to the number of 16S rRNA loci of the corresponding bacterial genome (i.e., 4 and 2 copies for *S. alvi* and *B. apis*, respectively). Reported copy number values were obtained from 3 independent experiments of 5 biological replicate each.

## Honey bee rearing and gut colonization

Microbiota-depleted bees *Apis mellifera carnica* were sourced from outdoor colonies kept at the University of Lausanne (VD, Switzerland), as previously described [58]. In summary,

newly emerged honey bees were obtained by transferring mature pupae (i.e., pigmented eyes and light grey cuticula) from brood frames to sterilized plastic boxes. Pupae were kept for 3 days at 35˚C with 75% humidity and emerging adult bees had access to a source of sterile 1:1 (w/v) sucrose water. A day prior, the hindguts of 2 newly emerged bees per pupae box were dissected and homogenized in 1 ml sterile 1X PBS to check for contaminated bees. For this, gut homogenates were plated onto NA, CBA, and MRSA solid media and incubated in normal atmosphere, microaerophilic and anaerobic environment, respectively. Pupae boxes for which bacterial growth was observed on the plates of the corresponding tested bees were discarded.

Gut monocolonization of the microbiota-depleted bees was carried out by individually feeding them 5 μl of solutions of *S. alvi* cells at an $OD_{600}$ of 0.1 resuspended in 1:1 (v/v) 1X PBS:sucrose water, representing an average of $1.9 \ 10^4 \pm 1.89 \ 10^3$ bacteria per inoculum. For colonization by *S. alvi* with the natural microbiota, gut homogenate stocks were obtained by combining in equal volumes the homogenized hindguts of 5 hive bees. Solutions used to feed the bees for colonization were then prepared by mixing *S. alvi* cells to a final $OD_{600}$ of 0.1 with 1:10 (v/v) diluted gut homogenate stock in 1:1 (v/v) 1X PBS:sucrose water. Colonized bees were kept at 32˚C with 75% humidity in sterile cup cages with added sterilized pollen and sucrose solution, the latter supplemented with 30 μg ml$^{-1}$ spectinomycin, when required. Honey bees monocolonized with the engineered *S. alvi* strain bearing the IPTG-inducible pAC17V5 plasmids were similarly fed with sterilized pollen and sucrose solution, but the latter was supplemented with 30 μg ml$^{-1}$ spectinomycin, 30 μg ml$^{-1}$ ampicillin, and 0.1 or 1 mM IPTG, as appropriate. Supplemented sucrose solutions (i.e., spectinomycin and IPTG) were replaced with freshly prepared tubes every 3 days throughout the experiments to ensure constant concentrations of the corresponding molecules.

## Bacterial load and plasmid maintenance assessment via fecal sampling

To sample feces from honey bees, colonized bees in cup cages were first stunned using $CO_2$ and immobilized at 4˚C on ice. Gentle pressure was applied by hand on the abdomens of asleep bees using a slight motion, starting from the base of the abdomen towards the stinger, until defecation occurred into the cap of sterile 1.5 ml Eppendorf tubes. Pressure applied on the abdomen of the bees was relieved 2 seconds after defecation, which resulted in about 4 μl of feces on average (**Fig 1B**). Samples were kept on ice until the end of the sampling procedure.

To assess bacterial load from the obtained feces, fecal matter was either directly serially diluted 1:10 (v/v) in sterile 1X PBS if liquid enough for pipetting or first resuspended in 4 μl of sterile 1X PBS prior to the serial dilution if too difficult to pipet. Diluted feces were then plated on TSA solid media supplemented with or without 60 μg ml$^{-1}$ spectinomycin, as appropriate. CFUs of the engineered *S. alvi* cells were then counted after 3 to 4 days incubation. For the assessment of plasmid maintenance, fluorescent CFUs were discriminated against nonfluorescent colonies and counted from images of the plates taken by EPI fluorescence with a Fusion FX (Vilber Lourmat) apparatus (high sensitivity, 8 aperture, F-740 filter). A median value of 69.5 colonies were counted and analyzed to obtain the plasmid maintenance percentage for each sample.

## Honey bee survival assays

To measure the impact of fecal extraction on honey bee's health, we reared 2 cohorts of honey bees, with one subjected to weekly feces extraction (as described above) and a second control group that was not manipulated. To account for cage effect, each cohort comprised 2 separate cages of at least 10 honey bees. For 22 days, survival of bees was monitored daily, during which occasions dead individuals were also removed.

## Gut and feces microscopy analysis

To prepare samples for microscopy, the hindguts of honey bees were dissected a week after inoculation with the engineered *S. alvi* and immediately transferred to 4% paraformaldehyde in PBS and fixed overnight at 4˚C with rotation. Fixed samples were then washed 3 times for 30 minutes in PBS. Hindguts were permeabilized and stained overnight in the dark at 4˚C with rotation, with 5 µg ml$^{-1}$ 4,6-diamidino-2-phenylindole (DAPI; ex. 359 nm–em. 457 nm) in 1% Triton X-100 in PBS. Excess dye was then washed from fixed samples with PBS, and the ileum of stained guts was dissected out, mounted in PBS, and overlaid with a coverslip (number #1.5). The laser scanning confocal microscopy was performed using an inverted Zeiss LSM 980 Airyscan 2 microscope (Carl Zeiss AG, Jena, Germany). Images were acquired with a 512 × 512 pixel format size (134.6873 × 134.6873 µm and depth of 1.8400 µm), an averaging of 4 images and a 63× oil objective.

For intensity measurement, all ileums were imaged with the same laser intensity and master gain on the detector. Three distinct sections per ileum were imaged. File names were randomized for blind intensity quantification. In Fiji, each image was split into DAPI and GFP channels, and the freeform tool was used to draw 3 densely colonized areas of each ileum section, based on the DAPI channel. Host cells were excluded from the area of interest. Selected areas were then copied to the corresponding images of the GFP channel. The intensity measurement used was the mean intensity per unit area of each specified region with the background subtracted. Normalized intensities of the 3 selected areas per image were averaged, resulting in a single value per ileum section. Finally, GFP intensities reported per gut were mean values of the averaged signals obtained from the corresponding 3 ileum sections imaged. To ensure differences in GFP fluorescence were due to IPTG rather than analytical biases, similar quantification of DAPI signal was performed on the same selected image areas, which showed no significant difference in DAPI signal between the different IPTG treatments (**S8 Fig**).

To quantify the GFP fluorescence of *S. alvi* from fecal samples, feces were first collected from honey bees colonized with the engineered bacteria a week after inoculation and immediately diluted 1:10 in sterile PBS. Before being imaged with the same microscopy settings described above for the honey bee guts, 5 to 10 µl of diluted feces was overlaid with a coverslip. Seven distinct areas per feces sample were imaged. Fluorescence quantification of cells was automated via a Fiji macro developed for this study (see Code availability section). Macro was run using the software Fiji (ImageJ2, version 2.9.0). Briefly, cells were counted and segmented based on a GFP fluorescence threshold, and the GFP signals of all cellular areas were averaged per image. Finally, the mean GFP intensities reported per bee (i.e., per fecal sample) were mean values of the averaged signals obtained from the corresponding 7 sample areas imaged.

## Supporting information

**S1 Data. Source data underlying plots shown in the study.**
(XLSX)

**S1 Fig. Bacterial load from feces is a proxy for levels of gut colonization.** Scatterplot shows linear regression of bacterial concentration values of engineered *S. alvi* found in matching samples of feces and gut homogenates (i.e., feces and gut were sourced from the same bee). Pearson correlation coefficient R and *p*-value are provided, for *n* = 22. The data underlying this Figure can be found in the **S1 Data** file, sheet "Supplementary Fig 1".
(PDF)

**S2 Fig. Plasmid maps of the broad-host range vectors developed in this study.** Broad-host-range replicons were sourced from the pSEVA1213S, pBMTBX-2, pME6012, and pDR401

plasmids (top row). The standard fragments bearing the antibiotic marker and fluorescent protein were obtained from the RSF1010-based vectors pBTK570 [28], pAC08, and pAC09 (left column). In those, RSF1010 was replaced with the different broad-host range replicons, resulting in the 11 new vectors shown.
(PDF)

**S3 Fig. Validation of qPCR primers and standard curves.** Primers specificity was confirmed by visualization of amplicons on an agarose gel (left panel) and generation of melting curves (middle panel). Standard curves were generated using serially diluted genomic DNA of **(a)** *S. alvi*, **(b)** *B. apis*, or **(c)** miniprep of pBTK570. The data underlying this Figure can be found in the **S1 Data** file, sheets "Supplementary Fig 3A," "Supplementary Fig 3B," and "Supplementary Fig 3C."
(PDF)

**S4 Fig. Characterization of functional broad-host range replicons in the honey bee gut symbiont *B. apis*. (a)** Broad-host range plasmids have different copy numbers in *B. apis*. Box plots show median values of plasmid copy numbers obtained by qPCR from 3 independent experiments with 5 biological replicate each (total *n* = 15). Median copy number are indicated with the corresponding box plots. **(b)** The difference in plasmid copy number results in different protein expression levels in *B. apis*. Graph shows mean of E2-crimson fluorescence ± standard deviations of 5 biological replicates. Each replicate represents the average fluorescence of at least 9,000 cells measured by flow cytometry. Plasmids used for panels **a** and **b** in *B. apis* were pBTK570, pAC06, pAC11, and pAC04, carrying the RSF1010, RK2, pTF-FC2, and pBBR1 origins of replication, respectively. **(c)** Some replicons are compatible and can be cotransformed in *B. apis*. Matrix table indicates compatible (green boxes with check mark) and incompatible (red boxes with cross mark) replicons. Vectors were found compatible upon their successful cotransformation by conjugation in *B. apis* cells. The data underlying this Figure can be found in the **S1 Data** file, sheets "Supplementary Fig 4A" and "Supplementary Fig 4B."
(PDF)

**S5 Fig. The plasmid pBTK552 is genetically unstable in our experimental conditions. (a)** Graph shows mean ± standard deviation percentage of bacterial population GFP positive. Three biological replicates were tested for each construct. Each replicate value is based on the average fluorescence of at least 9,000 *S. alvi* cells measured by flow cytometry, which were grown in 3 ml of TSB with (+) or without (−) IPTG for 3 days. As a reference, *S. alvi* bearing the pAC08 plasmid constitutively expressing GFP and *S. alvi* carrying our pAC17V5 dual-vector system were also analyzed. **(b)** The CP25 interregion gets deleted from pBTK552. Plasmid map of the pBTK552 vector (top panel) with a representative Sanger sequencing of a frequent deletion obtained in the unstable region (bottom panel) is shown. Dotted lines indicate the position of the deletion. The data underlying this Figure can be found in the **S1 Data** file, sheet "Supplementary Fig 5A."
(PDF)

**S6 Fig. Testing of different IPTG-inducible constructs in *S. alvi*. (a)** Maps of the IPTG-inducible plasmids built in this study. **(b)** *S. alvi* cells engineered with our inducible plasmids respond to IPTG exposure in vitro. Graph shows box plots representing median value of GFP fluorescence of 5 biological replicates for each construct tested. Each replicate value is based on the average fluorescence of at least 9,000 *S. alvi* cells measured by flow cytometry, which were grown in liquid with (+) or without (−) IPTG. As a reference, wild-type *S. alvi*, *S. alvi* bearing the pAC08 plasmid constitutively expressing GFP, and *S. alvi* carrying the previously

built pBTK552 vector [28] were also analyzed. Fold-changes of average fluorescence between uninduced and induced cells are indicated. The data underlying this Figure can be found in the **S1 Data** file, sheet "Supplementary Fig 6B."
(PDF)

**S7 Fig. The IPTG-inducible construct pAC17V3 allows honey bee and stingless bee gut symbionts to sense and respond to IPTG.** *Snodgrassella alvi* ESL0693, *Gilliamella apicola* wkB7, and *Gilliamella apis* ESL0169 cells engineered with the inducible plasmid pAC17V3 respond to IPTG exposure in vitro. Graphs show box plots representing median value of GFP fluorescence of 5 biological replicates for each condition. Each replicate value is based on the fluorescence of cells grown as bacterial lawns onto solid media (TSA) supplemented with (+) or without (−) 1 mM IPTG. Fold-changes of average fluorescence between uninduced and induced cells are indicated. Fluorescence was determined from images taken by EPI fluorescence with a Fusion FX (Vilber Lourmat) apparatus (16 aperture, F-740 filter) and identical exposure times between conditions. Analysis was performed with the software Fiji (ImageJ2, version 2.9.0). The intensity measurement used was the mean intensity per unit area of bacterial lawn with the cell and media autofluorescence subtracted. Autofluorescence values were obtained for each strain based on the intensity measured from bacterial lawns of the corresponding wild-type cells (i.e., not bearing the pAC17V3 plasmid). The data underlying this Figure can be found in the **S1 Data** file, sheet "Supplementary Fig 7."
(PDF)

**S8 Fig. DAPI measurements from gut tissues.** Graph shows box plots representing median value of DAPI fluorescence of *S. alvi* biofilms imaged from the gut of bees fed sugar water supplemented with either 0, 0.1, or 1 mM IPTG. Five bees were analyzed for each condition, and fluorescence values were averaged from 3 distinct sections of each gut. One-way ANOVA test, not significant (ns) with $q$-value $>$ 0.5. The data underlying this Figure can be found in the **S1 Data** file, sheet "Supplementary Fig 8."
(PDF)

**S9 Fig. Cell gating for flow cytometry analysis.** Graphs show pseudocolor plots used for gating of *S. alvi* cells. Quadrant limits were determined based on the measured fluorescence of reference cells bearing **(a)** the empty backbone pAC07 (no fluorescence), **(b)** pAC08 (GFP alone), or **(c)** pBTK570 (E2-crimson alone).
(PDF)

**S1 Table. Plasmids used in this study.**
(PDF)

**S2 Table. Primers used in this study.** Uppercase letters indicate priming sequence, and lowercase nucleotides show homology regions for Gibson assembly as primer overhangs.
(PDF)

**S1 Text. Fiji macro.**
(PDF)

# Acknowledgments

We thank Estelle Pignon, Emanuele Boni, and Morin Chhun for helpful scientific discussions throughout the project. We thank Paul Lachat for his support in the laboratory. We also acknowledge Théodora Steiner for her guidance in initiating the work with the bee gut symbionts. In addition, we thank Dr. Ismael Torres Romero for his assistance in writing the Fiji macro.

## Author Contributions

**Conceptualization:** Audam Chhun, Philipp Engel, Yolanda Schaerli.

**Data curation:** Audam Chhun.

**Formal analysis:** Audam Chhun, Silvia Moriano-Gutierrez.

**Funding acquisition:** Amélie Cabirol, Philipp Engel, Yolanda Schaerli.

**Investigation:** Audam Chhun, Silvia Moriano-Gutierrez, Florian Zoppi, Amélie Cabirol.

**Methodology:** Audam Chhun, Silvia Moriano-Gutierrez, Amélie Cabirol.

**Project administration:** Philipp Engel, Yolanda Schaerli.

**Software:** Audam Chhun.

**Supervision:** Audam Chhun, Philipp Engel, Yolanda Schaerli.

**Validation:** Audam Chhun.

**Visualization:** Audam Chhun.

**Writing – original draft:** Audam Chhun.

**Writing – review & editing:** Audam Chhun, Silvia Moriano-Gutierrez, Amélie Cabirol, Philipp Engel, Yolanda Schaerli.

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
