## [Editor Report · Decision Letter 0]

12 Jul 2023

Dear Dr Schaerli, 

Thank you for submitting your manuscript entitled "Engineering a symbiont as a biosensor for the honey bee gut environment" for consideration as a Methods and Resources Article by PLOS Biology. Please accept my apologies for the delay in getting back to you as we consulted with an academic editor about your submission.

Your manuscript has now been evaluated by the PLOS Biology editorial staff, as well as by an academic editor with relevant expertise, and I am writing to let you know that we would like to send your submission out for external peer review.

Once your full submission is complete, your paper will undergo a series of checks in preparation for peer review. After your manuscript has passed the checks it will be sent out for review. To provide the metadata for your submission, please Login to Editorial Manager (https://www.editorialmanager.com/pbiology) within two working days, i.e. by Jul 14 2023 11:59PM.

Kind regards,

Richard

Richard Hodge, PhD

rhodge@plos.org

PLOS

---

## [Decision Letter · Decision Letter 1]

16 Aug 2023

Dear Dr Schaerli,

Thank you for your patience while your manuscript "Engineering a symbiont as a biosensor for the honey bee gut environment" was peer-reviewed at PLOS Biology. Please accept my sincere apologies for the delays that you have experienced during the peer review process. Your manuscript has now been evaluated by the PLOS Biology editors, an Academic Editor with relevant expertise, and by three independent reviewers. 

In light of the reviews, which you will find at the end of this email, we would like to invite you to revise the work to thoroughly address the reviewers' reports.

As you will see below, Reviewers #1 and #3 are generally positive about the method and think that the IPTG biosensor will be useful for the field. Reviewer #2 raises concerns with the overall strength of the technical advance and notes that the impact of the work could be enhanced by further generalizing the method, such as demonstrating the utility of the biosensor in another bee-related bacterium or by developing another biosensor that can sense natural compounds. Reviewer #3 agreed with this comment during reviewer cross-commenting that we offer at the journal. After discussions with the Academic Editor, we feel that the development of a new biosensor is outside the scope for a revision, but we agree that incorporating data showing the application of the IPTG biosensor to other bee-related bacteria would strengthen the manuscript. 

Given the extent of revision needed, we cannot make a decision about publication until we have seen the revised manuscript and your response to the reviewers' comments. Your revised manuscript is likely to be sent for further evaluation by all or a subset of the reviewers.

**IMPORTANT - SUBMITTING YOUR REVISION**

*Re-submission Checklist*

*Published Peer Review*

*PLOS Data Policy*

*Blot and Gel Data Policy*

Sincerely,

Richard

Richard Hodge, PhD

rhodge@plos.org

REVIEWS:

Reviewer #1: The manuscript by Chhun and colleagues describes a new genetic toolkit for the honey-bee, allowing for approaches to study host-gut microbiota interactions. The authors also engineered the native bee gut bacterium Snodgrassella alvi as a biosensor for IPTG and showed that fluorescence readout can be measured in the gut tissues or non-invasively in the faeces. 

Overall, I think that this is an exciting and environmentally important model system that is currently underdeveloped. Tools such as these could open up a wide range of studies in terms of biological understanding and provide novel modalities for manipulation. I have some suggestions/comments for the improvement of the manuscript.

Line 32: There are also electronic auto sampling devices that can be used which may be termed non-invasive. I think these should be mentioned.

Figure 1d: Although the Kaplan-Meier test gives an insignificant p value from the curve it looks like there is an impact on fitness and perhaps that further sampling may be detrimental. Perhaps this could be commented on in the discussion, saying further study is warranted.

Line 201: You mention plasmid maintenance systems here and in the discussion. You know there is one on pTF-FC2. Are there others on the plasmids you are using? Could you consider adding maintenance systems to the other plasmids?

Reviewer #2 (Tae Seok Moon, signs review): In this work, the authors claim that they could measure fluorescence in the gut tissues or in the feces of honey bees by reprogramming a native bee gut bacterium as a biosensor for IPTG. It is an interesting paper with unusual environments and topics of interest (e.g., bee feces and symbionts). Conceptually, this work would be novel with interesting angles of research. Technically, however, the work is a bit simple as is; too simple to make a big impact or significant advances in synthetic biology. As is, the work would attract a limited number of readers. The following comments are given to make it stronger.

1] There are many practical biosensors that have been developed, mostly for applications in the mouse gut microbiome. Although this work focuses on honey bees' microbiota, it would be great for the authors to expand their discussion on biosensors developed for microbiome engineering and applications.

2] The IPTG sensor developed here might be novel in the specific species, but it is not practically interesting or useful, given the two-decade-old history of synthetic biology, generating multiple practical sensors. This work would have a bigger impact if the authors demonstrate the sensor's generalizability (e.g., demonstrating the functional IPTG sensor in another bee-related bacterium) or a more useful sensor for practical applications. Alternatively, they could develop another sensor in the same bacterium (e.g., sensors for anhydrotetracycline [aTc] although aTc is not interesting or useful, or other chemicals).

3] qPCR data reporting should follow generally accepted guidelines (i.e., MIQE). It does not look that MIQE has been followed.

4] Fig 2 can be a supplementary figure.

5] Fig 5. It looks that the sensor showed statistically significant sensing, but the fold change in response to different IPTG conditions (even at the highest concentration) is too small to be practical. Thus, along with my comments above, my enthusiasm for this work reduced, and this issue should be addressed.

Reviewer #3: This paper by Chhun et al. reports the development of molecular toolkits to genetically engineer the honey bee symbiont, Snodgrassella alvi, as a biosensor, using IPTG as a proof of concept. There is a need of using native symbionts for more accurate insights into the biology of host-associated tissues, since the majority of the reported biosensors rely on E. coli or a few other bacteria that would not readily persist in many non-model animal systems, including the honey bee. The current genetic toolkits which have been effectively used in the honey bee gut microbiome system are relatively large and complex, making them difficult to clone; furthermore, the lack of other replicons makes it difficult to build and use multiple constructs together. The authors results include a novel plasmid construct, pAC17V5, which is a circuit system composed of 2 plasmids, that allow S. alvi to exhibit a response to increasing doses of IPTG by reporting it as increasing GFP intensities.

The authors also propose a method for sampling and reporting bee gut contents via their faecal matter without the need to dissect the bees. The lack of established methodologies for faecal analysis in honeybees is a known problem, as honey bees do not typically defecate in laboratory conditions. 

Overall, the paper provides a promising approach to address the challenges associated with studying host-microbiota interactions in the context of honey bee gut microbiota, and a strong proof-of-concept as to using engineered microbes as biosensors in this environment. It is slightly disappointing that they did not extend their work towards the sensing of natural compounds or environmental conditions that occur in the gut. Doing so may have revealed some novel biology and would have been a valuable addition to the study. As it stands, this study is nonetheless an impressive paper focused on method development in the bee gut system. The tools described here will likely be used by many researchers in the future.

While the paper is generally well written and the results are exciting, there are certain areas that are lacking in detail or require clarification.

Line 98: How the bees were stunned should be mentioned here (CO2), in addition to in the methods.

Line 99: In figure 1b, the average is given as 4.35 ul. Looking at the supplementary data, it seems that the 4.8 ul value mentioned in the text here excludes samples with 0 ul values. The authors should be consistent and include the zero value samples in their statistical calculations, since they represent real measured replicates.

Figure 1d: How many times was this experiment replicated? Survival assays can suffer from high variance, especially in animal trials, and it would be better if these experiments were performed in triplicate.

Line 119: Is there a correlation between the amount of feces extracted from an individual and their survival probability?

Table 1: Please clearly define the criteria for a plasmid being considered "functional".

Lines 175-181: This hypothesis can likely be tested by plating of their experiment of panel 4b on selective (spectinomycin?) media in addition to nonselective media. This would allow visualization of differences between the absolute CFU counts of Ab+ and Ab- treatments, and can confirm plasmid loss during culture. 

Figure 4, caption title: Remove "in the honey bee gut.", since panel 4A is data from in vitro work.

Figure 4: It is not clear which plasmid versions are being tested here. In Fig. 2, there are plasmid variants constructed with combinations of GFP, Crimson, and ampR and specR. It seems from the text that Crimson-specR was used, but it should be made explicitly clear, and the rationale given for this choice versus the other possible combinations.

Figure 4d: For clarity, I suggest adding "-Ab" to the column with "+Gc", since I presume it is also without antibiotics.

Figure 4d: These samples were plated on selective media. There should have been controls to show that the +Gc microbiota do not have native resistance to spectinomycin at the concentrations used, which could confound the results. Even though the results for pVS1 suggest that the +Gc microbes are not resistant to their antibiotic, it is possible if each trial was done with a separate batch of gut inocula, there would be differences between the batches in terms of resistance.

Figure 4d: There are many zero values for the CFU counts, which require explanation. The median values for each treatment listed in the supplementary raw data file do not correspond to the indicated medians in the figure, nor do the interquartile ranges seem to make sense. It seems the authors have excluded the zero values from their analysis, and this should be clearly stated and justified. 

Figure 4: Analyses 4c and 4d seem to have been performed on the same samples. Therefore, it is odd that the number of samples (n's) analyzed in the panels are different. This should be explained. For instance, were the samples chosen for plating on non-selective media (Fig. 4c) a randomly chosen subset from 4d?

Figure 4: pBBR1 panel 4c -Ab treatment shows a drastic drop from day 7 to day 14; however, there are still many cells (around 10^3) from the day 14 sample as shown in the corresponding panel 4d -Ab treatment day 14. Do the authors imply that the lack of fluorescence is from the loss of the plasmid after growth on non-selective media, as in the case of pVS1? Also, it is not clear whether the zero values for plasmid maintenance in panel 4c is because they did not observe growth, or whether the colonies did not fluoresce. There should be mention of how many colonies were counted to obtain the percentages shown.

Lines 196-198: The assertion that pBBR-1 is lost quickly contrasts with line 220, where they say that this plasmid is stably maintained even without antibiotics.

Line 185: Please specify what is the CFU of the initial inoculum was, since this would help you interpret whether the bacteria are actively colonizing vs. persisting/dying over time.

Line 214: Specify what is the "selective solid media". Spectinomycin?

Line 250: Please provide more details as to the experiment where it was determined the pBTK552 plasmid had low stability. For instance, what was the time interval over which stability was tested?

Figure 5e and text Lines 604-609: It is unclear here exactly which types of images were taken for fluorescence quantification. Is the image in panel 5d an representative image? How big of an area was imaged (can you provide dimensions)? How big were the areas of host and bacterial cells selected for analysis? How many of each type of cell is expected to be included in such a selection?

Lines 466-482: A new electroporation method was developed, which is mentioned as a faster and simpler method to transform S. alvi. This is compared to the previous conjugation method. However, there are no data provided as to the transformation efficiency of this new electroporation method.

---

## [Editor Report · Decision Letter 2]

26 Jan 2024

Dear Dr Schaerli,

Thank you for the submission of your revised Research Article "Engineering a symbiont as a biosensor for the honey bee gut environment" for publication in PLOS Biology. On behalf of my colleagues and the Academic Editor, Baojun Wang, I'm pleased to say that we can in principle accept your manuscript for publication, provided you address any remaining formatting and reporting issues. These will be detailed in an email you should receive within 2-3 business days from our colleagues in the journal operations team; no action is required from you until then. Please note that we will not be able to formally accept your manuscript and schedule it for publication until you have completed any requested changes.

NOTE: I have asked my colleagues to include in that list of requests the following two editorial requests:

a) Please change your Title to "An engineered bacterial symbiont allows non-invasive biosensing of the honey bee gut environment"

b) Please cite the location of the data clearly in all relevant main and supplementary Figure legends; to do so, please re-name "source_data_revised" to "S1_Data" and write e.g. “The data underlying this Figure can be found in S1 Data” in each legend.

Sincerely, 

Roli Roberts

Roland G Roberts PhD

Senior Editor

PLOS Biology

rroberts@plos.org

on behalf of

Richard Hodge, PhD, 

Senior Editor

PLOS Biology

rhodge@plos.org